# Information-Theoretic Conditions for Chain-of-Thought Monitorability and Methods for Improving It

## Abstract

Chain-of-thought (CoT) monitors are LLM-based systems that analyze reasoning traces to detect when outputs may exhibit attributes of interest such as harmfulness, bias, or other properties. In this paper, we develop a rigorous information-theoretic framework to analyze the fundamental conditions that determine CoT monitorability. Our analysis establishes two necessary conditions for successful monitoring: (i) the CoT must encode sufficient information about the target attribute, and (ii) the monitor must be capable of extracting this information. We show that the success of CoT monitoring hinges on conditional mutual information between outputs and CoTs. We further demonstrate that CoT monitorability can be systematically improved through targeted training objectives. To this end, we propose two complementary approaches: (a) an oracle-based method that directly rewards the monitored model for producing CoTs that maximize monitor accuracy, and (b) a more practical, label-free approach that maximizes conditional mutual information between outputs and CoTs. Both methods significantly improve monitor accuracy while preventing CoT degeneration even when training against a monitor, thereby mitigating reward hacking even when the task reward is imperfectly specified.

## 1 Introduction

Modern large language models (LLMs) are increasingly trained to generate reasoning traces, also called chain-of-thought (CoT), before producing final outputs. This capability has enabled the development of CoT monitors – LLM-based systems that analyze these reasoning traces to detect attributes such as harmfulness or bias in the resulting outputs. By analyzing the CoT, monitors can theoretically identify problematic attributes *before* they manifest in the output, e.g, flawed reasoning patterns or attempts at deception. However, our understanding of when – and to what extent – CoT monitors can provide robust supervision remains limited. Challenges such as CoT obfuscation [Baker et al., 2025], steganographic reasoning [Stickland and Korbak, 2025], and general lack of faithfulness in CoT reasoning [Turpin et al., 2023, Lanham et al., 2023] may significantly constrain the practical effectiveness of CoT monitoring.

This works aims to further our understanding of the robustness and utility of CoT monitoring. Our contributions include: (1) a rigorous probabilistic framework through which we establish necessary information-theoretic conditions for monitorability, and analyze the gap between ideal and practical CoT monitors; and (2) a practical training objective that demonstrably improves CoT monitor accuracy while preventing CoT degeneration.

Submitted to 39th Conference on Neural Information Processing Systems (NeurIPS 2025). Do not distribute.

## 2 Background

**Entropy.** The Shannon entropy of a discrete random variable $A$, denoted $H(A)$, measures its average uncertainty and is defined as $H(A) = -\sum_a p(a) \log p(a)$. The conditional entropy, $H(A \mid B)$, measures the remaining uncertainty in $A$ after observing $B$.

**Mutual Information.** The mutual information (MI) between two random variables, $I(A; B)$, quantifies the reduction in uncertainty about one variable from knowing the other. It is defined as:

$$I(A; B) = H(A) - H(A \mid B). \tag{2.1}$$

MI is symmetric, so $I(A; B) = I(B; A)$. A value of $I(A; B) = 0$ implies that the variables are independent. The conditional mutual information, $I(A; B \mid C)$, measures the MI between $A$ and $B$ given a third variable $C$. A key identity we will use is the chain rule for mutual information:

$$I(A, B; C) = I(A; C) + I(B; C \mid A). \tag{2.2}$$

### 2.1 Related Works

**CoT Monitoring and Evasion.** Chain-of-thought (CoT) monitoring has been proposed as a mechanism for interpretable oversight of LLMs [Korbak et al., 2025]. However, recent work highlights its brittleness: CoT is not reliably faithful [Turpin et al., 2023, Lanham et al., 2023, Chen et al., 2025, Barez et al., 2025] – and may thus not reflect whether models are engaging in unwanted behavior – and models can potentially evade monitors through obfuscation [Baker et al., 2025] or steganography [Motwani et al., 2024, Mathew et al., 2024]. Emmons et al. [2025] show that monitoring is more effective on tasks that require CoT, while Stickland and Korbak [2025] explore generator-to-monitor transferability. These findings motivate a more formal understanding of when CoT monitoring succeeds.

**Information-Theoretic Approaches.** Our analysis builds on a growing body of work using information-theoretic tools for interpretability and control. Prior work has applied information-theoretic tools to interpretability and alignment [Xu et al., 2020, Belrose et al., 2025], and has analyzed CoT through information gain and bottleneck principles [Ton et al., 2025, Lei et al., 2025]. However, these approaches do not address monitorability explicitly.

**Training Strategies and Reward Hacking** Training models to remain monitorable is challenging: optimizing against monitors may induce obfuscation [Baker et al., 2025], echoing broader concerns about alignment faking and deceptive behaviors [Park et al., 2023, Scheurer et al., 2024, Greenblatt et al., 2024, Meinke et al., 2025] and multi-agent collusion [Motwani et al., 2024, Dou et al., 2025]. Our work introduces a label-free conditional mutual information objective that aims to preserve monitorability and mitigate reward hacking, complementing broader efforts to improve reasoning faithfulness [Paul et al., 2024] and robustness of RL training [Farquhar et al., 2025, Liu et al., 2025].

## 3 Information-Theoretic Analysis of CoT monitorability

In this section, we develop a formal framework to analyze when and why chain-of-thought (CoT) reasoning could be monitored effectively. We begin by formalizing the probabilistic setup and defining key concepts. We then identify necessary information-theoretic conditions for successful monitoring, showing that the conditional mutual information between CoTs and outputs, $I(Z; O|X)$, plays a critical role in determining monitorability. Finally, we analyze the gap between ideal (Bayes-optimal) monitors and practical, capacity-bounded monitors, quantifying how decoder limitations affect monitoring performance.

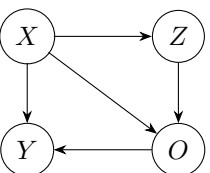

Figure 1: Graphical model of our Chain-of-Thought setup.

### 3.1 Setup and Definitions

We model a large language model as a policy, $\pi_\theta$, that generates text autoregressively. Our framework focuses on tasks where the model produces a Chain-of-Thought (CoT) before the final output. More

specifically, we assume the directed acyclic graph (DAG)

$$X \to Z, \qquad (X, Z) \to O, \qquad (X, O) \to Y,$$

where $X$ is the prompt, $Z$ is the chain-of-thought (CoT), $O$ is the model's final action/answer and $Y$ is the attribute of interest.

**Definition 3.1** (Action Monitor). *An* action *monitor is a stochastic mapping*

$$g \; : \; \mathcal{O} \times \mathcal{X} \; \longrightarrow \; \Delta(\mathcal{Y}), \qquad y \sim g(\,\cdot\mid O, X),$$

*where $\Delta(\mathcal{Y})$ denotes the probability simplex on $\mathcal{Y}$.*

**Definition 3.2** (CoT Monitor). *Similarly, CoT monitor is a stochastic mapping*

$$m_\psi \; : \; \mathcal{Z} \times \mathcal{X} \; \longrightarrow \; \Delta(\mathcal{Y}), \qquad y \sim m_\psi(\,\cdot\mid Z, X).$$

For both monitors, point predictions could be attained by performing argmax over them, e.g.,

$$y \; = \; \underset{Y \in \mathcal{Y}}{\operatorname{argmax}}\, m_\psi(Y \mid Z, X).$$

**Definition 3.3** (Bayes–Optimal CoT Monitor). *Let $\pi(O \mid Z, X)$ be the model's conditional distribution over final actions and let $g(Y \mid O, X)$ be the stochastic action-monitor defined above. The ground truth distribution of the attribute $Y$ given the prompt $x$ and chain-of-thought $z$ can be defined as*

$$m_g^\pi(Y = y \mid Z, X) = \mathbb{E}_{o \sim \pi(O \mid Z, X)}[g(Y = y \mid O = o, X)].$$

*Under the 0–1 scoring rule, the* Bayes–optimal CoT monitor *is the classifier that outputs the MAP label*

$$\hat{y} \; = \; m^\star(z, x) \; := \; \underset{y \in \mathcal{Y}}{\operatorname{argmax}}\, m_g^\pi(Y = y \mid Z = z, X = x).$$

**Definition 3.4** ($\alpha$-CoT monitorable task). *For $\alpha \in [0, 1]$, we say an attribute $Y$ is $\alpha$-CoT-monitorable with respect to policy $\pi$ if the Bayes-optimal CoT monitor $m^\star(z, x)$ has accuracy at least $\alpha$.*

Note that the performance of any practical CoT monitor is bounded above by the Bayes-optimal CoT monitor, whose performance in turn is bounded by the action monitor $g$. We emphasize that the CoT-monitorability of an attribute is intrinsically linked to the policy $\pi$ that generates the CoT being monitored. There are two natural corollaries of this fact: (a) an attribute $Y$ being monitorable w.r.t. some policy $\pi_1$ is no guarantee that the same attribute would be monitorable for some other policy $\pi_2$. (b) monitorability of an attribute $Y$ may be improved through changing the distribution of $Z$ induced by the policy, for example, by training $\pi$ to produce more informative chain-of-thought reasoning.

Due to space limitations, we sometimes omit proof from the main text, and defer them to the Appendix A.

## 3.2 Necessary Conditions for CoT Monitorability

In this section, we investigate the necessary information-theoretic conditions that determine when a CoT may be monitorable for some attribute of interest $Y$. Specifically, we analyze under what properties of the CoT and the policy $\pi$, a task could be considered $\alpha$-CoT monitorable.

Our analysis yields two main insights. Firstly, we show that conditional mutual information between CoT and the attribute of interest $I(Y; Z \mid X)$ gives the ceiling on the performance of *any* CoT monitor. Next, we show that positivity of mutual information between CoT and actions $I(O; Z \mid X)$ is a necessary condition for monitorability of any attribute. However, it is not a sufficient condition, and despite positive $I(O; Z \mid X)$, there may exist attributes which may be difficult to monitor.

**Proposition 3.5** (Necessary MI threshold for $\alpha$-monitorability). *Let $Y$ take values in a finite label set $\mathcal{Y}$ with $\kappa = |\mathcal{Y}| \geq 2$. For $\alpha \in [1/\kappa, 1]$, define*

$$T_\alpha \; := \; h_b(1 - \alpha) \; + \; (1 - \alpha) \log(\kappa - 1), \qquad h_b(u) = -u \log u - (1 - u) \log(1 - u).$$

*If there exists a (possibly stochastic) monitor $m_\psi(Y \mid Z, X)$ whose induced 0–1 accuracy is at least $\alpha$ under the data-generating distribution, then*

$$I(Y; Z \mid X) \; \geq \; H(Y \mid X) \; - \; T_\alpha. \tag{3.1}$$

The above proposition formalizes the intuition that a CoT-monitor can only successfully predict some attribute of interest $Y$ of the output from CoT to the extent that CoT encodes information about this attribute. Two important caveats apply: (1) it establishes existence of some monitor achieving accuracy $\alpha$ when $I(Y;Z|X)$ is sufficiently high, but does not guarantee that any specific monitor class (e.g., prompted LLMs) will attain this accuracy; (2) even with high mutual information, practical monitors may fail if the information is encoded in ways that resist extraction (e.g., steganographically). We address these limitations in our analysis of practical monitors in Section 3.3.

**Do CoTs that matter for the task also help monitoring?** Emmons et al. [2025] empirically demonstrate that tasks requiring explicit CoT reasoning are more amenable to CoT monitoring. Indeed, our theoretical analysis confirms this observation, by showing that the necessity of CoT reasoning for task completion is a necessary precondition for the monitorability of any attribute $Y$ of the final output. However, it is not a sufficient condition.

**Lemma 3.6** (MI decomposition). *Let $Y = g(O, X)$ for the stochastic map $g : O \times \mathcal{X} \to \mathcal{Y}$. Then*

$$I(O; Z \mid X) = I(Y; Z \mid X) + I(O; Z \mid Y, X). \tag{3.2}$$

*Proof.* Apply the chain rule two ways: $I(O, Y; Z \mid X) = I(Y; Z \mid X) + I(O; Z \mid Y, X) = I(O; Z \mid X) + I(Y; Z \mid O, X)$. Note that by assumption we have $Y \perp\!\!\!\perp Z \mid O, X$, which means $I(Y; Z \mid O, X) = 0$, which in turn gives (3.2). $\square$

**Proposition 3.7** (Necessity of CoT reasoning is a necessary but not sufficient condition for monitorability). *Assume $Y = g(O, X)$ where $g : O \times \mathcal{X} \to \mathcal{Y}$ is a deterministic map.*

1. *(Necessity) For any attribute $Y$, if a monitor $m_\psi(Z, X) \in M$ with non-trivial accuracy exists then $I(O; Z \mid X) > 0$.*

2. *(Not sufficiency) $I(O; Z \mid X) > 0$ does* not *imply there exists a monitor $m_\psi(Z, X) \in M$ with non-trivial accuracy.*

*Proof. Necessity:* By Proposition 3.5, existence of a monitor with non-trivial accuracy implies $I(Y; Z \mid X) > 0$. From Lemma 3.6 we have $I(O; Z \mid X) \geq I(Y; Z \mid X)$, so $I(Y; Z \mid X) > 0 \Rightarrow I(O; Z \mid X) > 0$.
*Not Sufficiency:* By Lemma 3.6, we have $I(O; Z \mid X) = I(Y; Z \mid X) + I(O; Z \mid Y, X)$. Because both terms on RHS are nonnegative, it is possible that $I(O; Z \mid Y, X) > 0$ while $I(Y; Z \mid X) = 0$, yielding $I(O; Z \mid X) > 0$. Because $I(Y; Z \mid X) = 0$, then by Proposition 3.5 we have that no CoT monitor with non-trivial accuracy exists. Because $I(Y; Z \mid X) = 0$, by the contrapositive of Proposition 3.5 we have that no CoT monitor with non-trivial accuracy exists. $\square$

### 3.3 Information Gap Between Practical Monitor and The Ideal Monitor

The analysis in the previous section considers Bayes optimal monitor that uses the ground truth decoder distribution $m_g^\pi$, and establishes when such a monitor might be realizable. However, in practice, CoT monitors are (prompted or fine-tuned) LLMs with finite capacity and thus there may be a decoder mismatch – the decoder $q(Y|Z, X)$ induced by the monitor LLM may be imperfect and may not match well with the Bayes optimal decoder $m^\star$. We make the loss of performance due to this decoder mismatch precise through the following proposition:

**Proposition 3.8** (Information gap.). *Let $\pi$ be the (true) distribution over $(Y, Z, X)$. Let the CoT monitor be a stochastic decoder $m_\psi(Y \mid Z, X)$ and define*

$$I_{m_\psi}(Y; Z \mid X) := \mathbb{E}_{\pi(Y,Z,X)}\left[\log \frac{m_\psi(Y|Z,X)}{m_\psi(Y|X)}\right], \quad m_\psi(Y \mid X) := \mathbb{E}_{Z \sim \pi(Z|X)}\left[m_\psi(Y \mid Z, X)\right].$$

*Then* $I(Y; Z \mid X) \geq I_{m_\psi}(Y; Z \mid X)$.

**Corollary 3.9** (The information gap is minimized by using the data generating distribution as the decoder). *$I(Y; Z \mid X) = I_{m_\psi}(Y; Z \mid X)$ if and only if $m_\psi(Y \mid Z, X) = \pi(Y \mid X, Z) = m_g^\pi(Y|X, Z)$.*

### 3.4 Key Takeaways

Our information–theoretic view highlights two distinct failure modes for CoT monitoring:

- **Encoding bottleneck.** If the CoT carries *no* information about the model's action, $I(O; Z \mid X) = 0$, then it also carries no information about any attribute of that action, $I(Y; Z \mid X) = 0$. In this regime no CoT monitor with non-trivial accuracy is realizable.

- **Decoder limitation.** When $I(O; Z \mid X) > 0$, the information may exist but a practical monitor $m_\psi(y \mid x, z)$ might fail due to its inability to extract this information.

# 4 Can CoT Monitorability be Improved?

Prior works, such as Baker et al. [2025], have focused on how CoT monitorability could be preserved. We go a step further, and consider what can be done to improve CoT monitorability of a task. Our analysis demonstrates that targeted training objectives can substantially improve the information content of CoTs, making them more amenable to monitoring.

We explore two primary approaches. First, we introduce an *oracle-based* method where the model is rewarded for producing CoTs that maximize the accuracy of a fixed, external monitor. While this provides a conceptual benchmark, its practical application is limited by the need for a reliable oracle to provide ground-truth labels.

To address this limitation, we propose a practical, *label-free* proxy objective: maximizing the conditional mutual information $I(O; Z \mid X)$ between the model's actions and its CoT. This objective encourages the model to generate reasoning traces that are inherently informative about its final decision, preventing the CoT from becoming decoupled or obfuscated. We show how this objective can be efficiently estimated and integrated into a reinforcement learning framework.

## 4.1 Oracle Monitor-in-the-Loop RL

As a first approach, we consider a setup with a fixed, pre-trained LLM CoT monitor $m_\psi$. We only update the policy $\pi_\theta$ of the monitored model. Let $g$ denote the oracle action monitor, then given a trajectory $(y, o, z, x)$, we define a binary oracle reward signal:

$$R_{\text{Oracle}}(o, z, x) := \begin{cases} +1, & \text{if } m_\psi(z, x) = g(o, x), \\ -1, & \text{otherwise.} \end{cases} \tag{4.1}$$

This signal directly rewards the model for generating a CoT $z$ that leads the monitor to the correct conclusion $m_\psi(z, x) = g(o, x) = y$. The total reward function is an augmentation of the original task reward $R_{\text{Task}}(o)$:

$$R_{\text{overall}} = R_{\text{Task}}(o) + \mu\, R_{\text{Oracle}}(o, z, x), \tag{4.2}$$

where $\mu \geq 0$ is a hyperparameter that controls the strength of the oracle feedback.

With a fixed monitor $m$, maximizing $\mathbb{E}[R_{\text{Oracle}}(O, Z, X)]$ encourages $\pi_\theta$ to produce CoTs that are more easily decodable by that specific monitor, indirectly maximizing $I_{m_\psi}(Y; Z|X)$. However, this method's reliance on an external oracle for labels $Y$ makes it costly and impractical in real-world scenarios, motivating the need for a label-free alternative.

## 4.2 A Label-Free Proxy: Maximizing $I(O; Z \mid X)$

To overcome the need for an oracle, we propose maximizing the conditional mutual information $I(O; Z \mid X)$ as a proxy objective. This encourages a strong coupling between the CoT and the final action, which is a prerequisite for monitorability (see Proposition 3.7).

The relationship between our proxy and the true monitorability objective, $I(Y; Z \mid X)$, is captured by the chain rule for mutual information (as in Lemma 3.6):

$$I(O; Z \mid X) = I(Y; Z \mid X) + I(O; Z \mid X, Y). \tag{4.3}$$

The term $I(O; Z \mid X, Y)$ represents the gap between the true and the proxy objective. This represents information that is captured by the proxy but is irrelevant to the attribute being monitored – such as stylistic variations in the outputs. Crucially, this term is bounded by the conditional entropy of the CoT and of the output: $H(Z \mid X, Y) \geq I(O; Z \mid X, Y) \leq H(O \mid X, Y)$. Thus, we have the following inequalities for our proxy objective:

$$I(O; Z \mid X) \leq I(Y; Z \mid X) + H(Z \mid X, Y),$$

$$I(O; Z \mid X) \leq I(Y; Z \mid X) + H(O \mid X, Y).$$

This inequality guarantees that an increase in our proxy objective, $I(O; Z \mid X)$, will lead to an increase in the true monitorability objective, $I(Y; Z \mid X)$, provided that $H(Z \mid Y, X)$ or $H(O \mid Y, X)$ is small. For a given attribute $Y$, $H(Z \mid X, Y)$ quantifies the intrinsic CoT diversity, while $H(O \mid X, Y)$ quantifies the intrinsic output diversity. In both cases, we expect smaller values for coarse-grained, semantic attributes that significantly constrain the LLM's generations (e.g., '*Is the model hard-coding the values of unit tests?*', compared to more fine-grained attributes (e.g., '*Does the generation contain the letter* e*?*').

### 4.2.1 Estimation and Practical Optimization of $I(O; Z \mid X)$

To use the conditional mutual information $I(O; Z \mid X)$ as a practical reward signal in a reinforcement learning setup, we require a per-rollout point estimator. For a rollout $i$ that produces an output $o_i$ and a CoT $z_i$ from a prompt $x_i$, we estimate the mutual information as:

$$\hat{I}_i = \underbrace{\log \pi_\theta(o_i \mid z_i, x_i)}_{\text{Output log-likelihood given CoT}} - \underbrace{\log \hat{\pi}_\theta(o_i \mid x_i)}_{\text{Estimated marginal log-likelihood}} \tag{4.4}$$

We leverage the structure of Group-based Reward Policy Optimization (GRPO), which provides multiple rollouts for each prompt. This allows for an efficient Monte Carlo approximation of the marginal term, $\hat{\pi}_\theta(o_i \mid x_i)$. For a prompt $x_i$, we use the corresponding group of CoTs, $\mathcal{G}(x_i) = \{z_k\}$, to estimate the marginal probability as:

$$\hat{\pi}_\theta(o_i \mid x_i) = \frac{1}{|\mathcal{G}(x_i)|} \sum_{z_k \in \mathcal{G}(x_i)} \pi_\theta(o_i \mid z_k, x_i). \tag{4.5}$$

This term represents the average likelihood of the output $o_i$ over all the different reasoning paths generated for the same prompt.

A naive maximization of $\hat{I}_i$ could encourage the model to generate unnecessarily long outputs to artificially inflate the log-probabilities. To counteract this, we normalize the estimate by the length of the output, $\text{len}(o_i)$, though we omit it from the notation for brevity. Furthermore, since mutual information is unbounded (for our purposes), directly using this normalized value as a reward could destabilize training by overpowering other objectives, such as the task reward $R_{\text{Task}}$. To ensure a balanced optimization, we bound $\hat{I}_i$ by applying some suitable transformation $f(\cdot)$. In our experiments, we use $f(a) = \tanh(2a)$ This gives us our final mutual information reward component:

$$R_{\text{MI}}(o_i, z_i, x_i) = f(\hat{I}_i) = \tanh(2\hat{I}_i). \tag{4.6}$$

The final reward function for training the policy is then a weighted combination of the task reward and our MI reward:

$$R_{\text{overall}} = R_{\text{Task}}(o_i) + \lambda_{\text{MI}} \cdot R_{\text{MI}}(o_i, z_i, x_i), \tag{4.7}$$

where $\lambda_{\text{MI}}$ is a hyperparameter that controls the strength of the monitorability objective.

## 5 Experiments and Results

In this section, we empirically validate our theoretical framework and demonstrate the effectiveness of our proposed approaches for improving CoT monitorability. We first describe our experimental setup, including models, datasets, and evaluation metrics, followed by a comprehensive analysis of our results.

### 5.1 Experimental Setup

**Environment:** Our primary experimental environment is inspired by the honeypot environment of Baker et al. [2025]. Specifically, we create a coding environment where the RL agent could get high reward either by implementing some sort of shortcut solution (e.g., hardcoding answers to known test cases) or a general solution. Our environment is based on the Mostly Basic Python Problems (MBPP) dataset [Austin et al., 2021]. This dataset contains 974 short Python programming tasks designed for entry-level programmers, each consisting of a natural language description, a reference solution, and three test cases to verify functional correctness.

In our setup, we sample a rollout by prompting an LLM to write code to solve a given Python programming problem described in natural language. Within this prompt, we also include two test-cases (henceforth referred to as 'revealed' test cases). We hold out a third test case to evaluate whether the model is writing code specific to the two revealed test cases or a general solution for the given prompt. The test reward is based solely on whether the model's written code passes the revealed test cases – +1 if it passes, 0 otherwise. As such, this environment provides an option to the model to attain high-reward by writing code that passes only the two revealed test-cases.

**Models:** We perform reinforcement learning training on Qwen2.5-3B-Instruct and Qwen2.5-7B-Instruct models; these are our $\pi_\theta$. To ensure our models explore both general and "hacky" solutions (code that only passes the revealed test cases, e.g., by hardcoding values), we performed a light round of supervised fine-tuning on a small dataset of code examples of test-hacking before performing reinforcement learning training. See Appendix B.1 for details.

**Monitor:** We use Qwen2.5-7B-Instruct as our monitor model, $m_\psi$. The monitor observes the prompt and the CoT of the monitored model, $\pi_\theta$, then outputs a binary decision regarding whether the model is implementing a true solution or a hacky solution. The monitoring reward $R_{\text{Monitor}}$ for $\pi_\theta$ is $+1$ when the monitor determines the model is not hacking, and $-1$ when it detects hacking. We provide the prompt that we use for the monitor model Appendix B.3.

**Training Details:** We use the Group Relative Policy Optimization (GRPO) algorithm [Shao et al., 2024] for RL training. The same hyperparameters were used for all experiments; we provide them them in Appendix B.2.

To establish a clear delineation between chain-of-thought reasoning and final outputs, we use a structural formatting reward in all the experiments. This reward incentivizes models to organize their responses into distinct XML-tagged sections: a `<think>...</think>` block containing the reasoning process (corresponding to CoT $z$) and a `...` block containing the final solution (corresponding to output $o$). This reward is quite quickly satisfied by LLMs in the training, and has no bearing on the results, hence we omit explicit mention of this reward in subsequent discussion.

**Evaluation Metrics:** We evaluate our approach using four key metrics:

- **Mutual Information (MI):** We measure $I(O; Z|X)$ between the model's outputs and CoTs to quantify the information content in the reasoning traces.
- **Hacking Rate:** The percentage of samples for which the code written by the model passes only the revealed test cases but fails the held-out test case.
- **Monitor Accuracy:** The accuracy of the monitor in correctly identifying whether the model is implementing a hacky solution.
- **True Pass Rate:** The percentage of samples for which the code written by the model passes all tests – both the revealed tests and the held out test. This can be looked at as a measure of true model performance.

## 5.2 Results and Analysis

In Figure 2 we show results for 3B and 7B models from Qwen-2.5 family trained under four different reward configurations. To make it easier to decipher the general trend, we plot using exponential moving average with $\alpha = 0.25$.

**Mutual information reward improves CoT-monitorability:** Figure 2 shows the evolution of MI ($I(O; Z|X)$) during training. Notably mutual information deteriorates throughout the RL training for models optimizing $R_{\text{Task}}$ and $R_{\text{Task}} + R_{\text{Monitor}}$. This decrease in MI for $R_{\text{Task}} + R_{\text{Monitor}}$ is accompanied by a decrease in monitor accuracy, indicating the obfuscation behavior due to optimizing against CoT monitor that was also observed by Baker et al. [2025].

Critically, both our proposed approaches – oracle reward and MI reward – successfully maintain higher MI values throughout training, and maintain high monitor accuracy. This confirms our theoretical

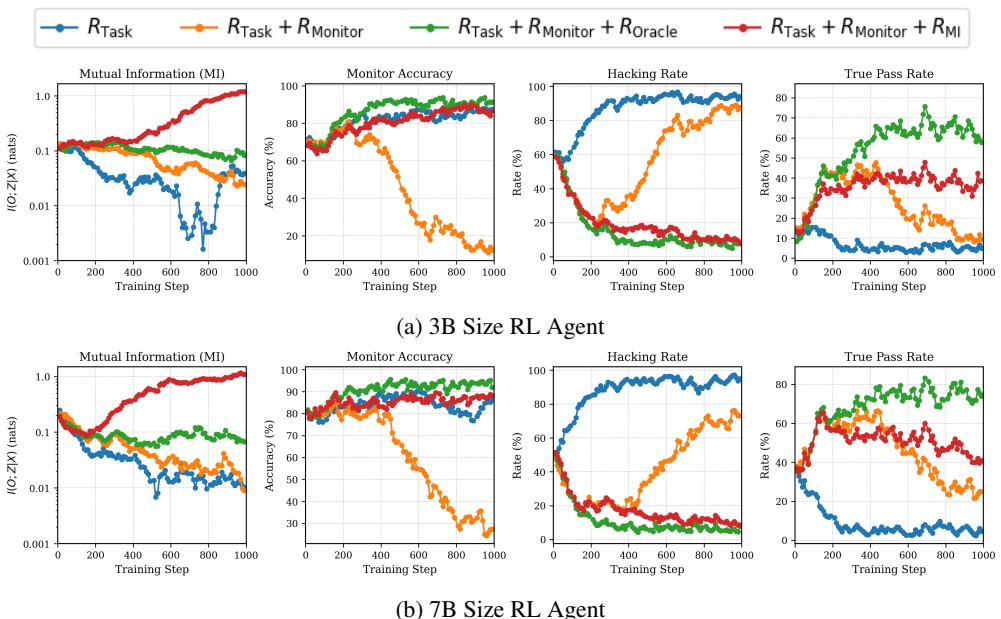

(a) 3B Size RL Agent

(b) 7B Size RL Agent

Figure 2: Comparison of (i) Mutual Information $I(O; Z|X)$, (ii) Monitor Accuracy, (iii) Hacking Rate, and (iv) Monitor Accuracy across training steps for the four different reward configurations.

prediction that maximizing $I(O; Z|X)$ is an effective strategy for preserving CoT monitorability, even when optimization pressure is being applied to CoT.

**Reward hacking and training against CoT-monitor:**  As shown in Figure 2, the model trained against a vulnerable task reward learns to hack it, and frequently produces code that only passes the revealed test cases. The model trained to optimize $R_{\text{Task}} + R_{\text{Monitor}}$ does initially learn to hack less but once it learns that it can evade the monitor via obfuscating its CoT, it reverts back to the hacking behavior while simultaneously improving on fooling the CoT monitor.

However, both models that additionally also optimize either $R_{\text{Oracle}}$ or $R_{\text{MI}}$ do successfully unlearn the hacking behavior and learn to not reward hack, but write true solutions to pass the revealed test cases. Intuitively, these terms make it difficult for the model to obfuscate its CoT which in turn means that CoT monitoring remains effective, even if it is an optimization target.

## 6  Discussion & Limitations

We assume a specific causal model. While we expect this causal model to be quite general and representative of many practical cases of interest, this does not apply in cases where the goal of CoT monitoring is not to predict the properties of the final action but rather the properties of the computational process, or the model, itself. For example, Baker et al. [2025] use CoT monitor to monitor for situational awareness which is a property of the model itself, and not an attribute of the model, and thus, falls outside this causal framework. Secondly, our theoretical analysis primarily establishes necessary conditions for successful CoT monitoring rather than sufficient ones. Finally, our validation of our proposed training objectives is currently limited.

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

# Appendix

## A Proofs

**Lemma A.1** (Fano's Inequality)**.** *Let $Y$ take values in a finite set $\mathcal{Y}$ with $|\mathcal{Y}| = \kappa \geq 2$. Let $(Z, X)$ be arbitrary side information and let $\hat{Y} = \hat{Y}(Z, X)$ be any estimator of $Y$. Denote its error probability by $P_{\mathrm{err}} = \Pr[\hat{Y} \neq Y]$. Then*

$$H(Y \mid Z, X) \ \leq \ h_b(P_{\mathrm{err}}) \ + \ P_{\mathrm{err}} \log(\kappa - 1), \qquad h_b(u) = -u \log u - (1 - u) \log(1 - u).$$

*In particular, the bound holds with $P_{\mathrm{err}} = P_{\mathrm{err}}^{\star}$ for the Bayes-optimal estimator.*

**Proposition A.2** (Necessary MI threshold for $\alpha$-monitorability)**.** *Let $Y$ take values in a finite label set $\mathcal{Y}$ with $\kappa = |\mathcal{Y}| \geq 2$. For $\alpha \in [1/\kappa, 1]$, define*

$$T_\alpha \ := \ h_b(1 - \alpha) \ + \ (1 - \alpha) \log(\kappa - 1), \qquad h_b(u) = -u \log u - (1 - u) \log(1 - u).$$

*If there exists a (possibly stochastic) monitor $m_\psi(Y \mid Z, X)$ whose induced 0–1 accuracy is at least $\alpha$ under the data-generating distribution, then*

$$I(Y; Z \mid X) \ \geq \ H(Y \mid X) \ - \ T_\alpha.$$

*Proof.* Let $m_\psi(Y \mid Z, X)$ be a (possibly stochastic) monitor whose induced 0–1 accuracy is at least $\alpha$. Let's denote the true value of the attribute be $y^\star$. Form a point estimator by taking the MAP decision

$$\hat{y} = \ \arg\max_{y \in \mathcal{Y}} m_\psi(y \mid Z, X),$$

breaking ties arbitrarily. ince MAP is at least as accurate (in 0–1 loss) as sampling from $m_\psi$, we have

$$P_{\mathrm{err}} \ := \ \Pr[\hat{y} \neq y_{\mathrm{true}}] \ \leq \ 1 - \alpha,$$

where $y_{\mathrm{true}}$ is the ground truth label.

By Lemma A.1 (i.e., Fano's inequality), we have

$$H(Y \mid Z, X) \ \leq \ h_b(P_{\mathrm{err}}) \ + \ P_{\mathrm{err}} \log(\kappa - 1)$$
$$H(Y \mid Z, X) \ \leq \ h_b(1 - \alpha) \ + \ (1 - \alpha) \log(\kappa - 1)$$
$$H(Y \mid Z, X) \leq T_\alpha.$$

By using the definition of mutual information and plugging in the above inequality, we get the desired result.

$$I(Y; Z \mid X) \ = \ H(Y \mid X) \ - \ H(Y \mid Z, X)$$
$$I(Y; Z \mid X) \geq \ H(Y \mid X) \ - \ T_\alpha.$$

$\square$

**Proposition A.3** (Information gap.)**.** *Let $\pi$ be the (true) distribution over $(X, Z, Y)$. Let the CoT monitor be a stochastic decoder $m_\psi(Y \mid Z, X)$ and define its $X$-marginal*

$$m_\psi(Y \mid X) \ := \ \mathbb{E}_{Z \sim \pi(Z|X)}\big[ m_\psi(Y \mid Z, X) \big],$$

*and the monitor MI*

$$I_{m_\psi}(Y; Z \mid X) \ := \ \mathbb{E}_{\pi(X, Z, Y)}\left[ \log \frac{m_\psi(Y \mid Z, X)}{m_\psi(Y \mid X)} \right].$$

*Then*

$$I(Y; Z \mid X) \ = \ I_{m_\psi}(Y; Z \mid X) \ + \ \mathbb{E}_{\pi(X, Z)}\Big[ D_{\mathrm{KL}}\big( \pi(Y \mid X, Z) \,\|\, m_\psi(Y \mid Z, X) \big) \Big]$$
$$- \ \mathbb{E}_{\pi(X)}\Big[ D_{\mathrm{KL}}\big( \pi(Y \mid X) \,\|\, m_\psi(Y \mid X) \big) \Big], \tag{A.1}$$

*and, in particular,*

$$I(Y; Z \mid X) \ \geq \ I_{m_\psi}(Y; Z \mid X). \tag{A.2}$$

*Proof.* Let $\pi$ denote the true data–generating distribution over $(X, Z, Y)$. Let $m_\psi(Y \mid X, Z)$ be any monitor (e.g., $m_\psi(Y \mid X, Z) = m_\psi(Y \mid Z, X)$), and define

$$m_\psi(Y \mid X) \ := \ \mathbb{E}_{Z \sim \pi(Z|X)}\big[ m_\psi(Y \mid X, Z) \big].$$

**Step 1: Start from the definition of conditional MI.**

$$I(Y; Z \mid X) \;=\; \mathbb{E}_\pi(X, Z, Y)\left[\log \frac{\pi(Y \mid X, Z)}{\pi(Y \mid X)}\right].$$

**Step 2: Multiply and divide inside the logarithm by** $m_\psi(Y \mid X, Z)$ **and** $m_\psi(Y \mid X)$ **and rearrange terms.**

$$I(Y; Z \mid X) \;=\; \mathbb{E}_\pi(X, Z, Y)\left[\log \left(\frac{\pi(Y \mid X, Z)}{\pi(Y \mid X)} \cdot \frac{m_\psi(Y \mid X, Z)}{m_\psi(Y \mid X, Z)} \cdot \frac{m_\psi(Y \mid X)}{m_\psi(Y \mid X)}\right)\right].$$

Rearranging terms and taking expectations under $\pi(X, Z, Y)$ gives

$$I(Y; Z \mid X) = \mathbb{E}_{\pi(X,Z,Y)}\left[\log \frac{m_\psi(Y \mid X, Z)}{m_\psi(Y \mid X)}\right] \;+\; \mathbb{E}_{\pi(X,Z,Y)}\left[\log \frac{\pi(Y \mid X, Z)}{m_\psi(Y \mid X, Z)}\right] \;+\; \mathbb{E}_{\pi(X,Y)}\left[\log \frac{m_\psi(Y \mid X)}{\pi(Y \mid X)}\right].$$

$$= I_{m_\psi}(Y; Z \mid X) \;+\; \mathbb{E}_{\pi(X,Z)}\Big[D_{\mathrm{KL}}\big(\pi(Y \mid X, Z) \,\|\, m_\psi(Y \mid X, Z)\big)\Big] \;-\; \mathbb{E}_{\pi(X)}\Big[D_{\mathrm{KL}}\big(\pi(Y \mid X) \,\|\, m_\psi(Y \mid X)\big)\Big]$$

This yields Equation A.1 completing the first part of the proof.

**Step 3: Conclude the lower bound** $I(Y; Z \mid X) \geq I_{m_\psi}(Y; Z \mid X)$**.** For each fixed $x$, by the log–sum inequality (equivalently, convexity of KL),

$$D_{\mathrm{KL}}\big(\pi(Y \mid X{=}x) \,\|\, m_\psi(Y \mid X{=}x)\big) \;\leq\; \mathbb{E}_{Z \sim \pi(Z|X=x)}\Big[D_{\mathrm{KL}}\big(\pi(Y \mid x, Z) \,\|\, m_\psi(Y \mid x, Z)\big)\Big].$$

Averaging over $x \sim \pi(X)$, and with slight abuse of notation we have

$$\mathbb{E}_{\pi(X)}\left[D_{\mathrm{KL}}\big(\pi(Y \mid X) \,\|\, m_\psi(Y \mid X)\big)\right] \;\leq\; \mathbb{E}_{\pi(Z,X)}\Big[D_{\mathrm{KL}}\big(\pi(Y \mid X, Z) \,\|\, m_\psi(Y \mid X, Z)\big)\Big].$$

$$0 \;\leq\; \mathbb{E}_{\pi(Z,X)}\Big[D_{\mathrm{KL}}\big(\pi(Y \mid X, Z) \,\|\, m_\psi(Y \mid X, Z)\big)\Big] - \mathbb{E}_{\pi(X)}\left[D_{\mathrm{KL}}\big(\pi(Y \mid X) \,\|\, m_\psi(Y \mid X)\big)\right].$$

This shows that the difference of the two KL terms in the identity is nonnegative, hence

$$I(Y; Z \mid X) \;\geq\; I_{m_\psi}(Y; Z \mid X).$$

$\square$

**Corollary A.4** (The information gap is minimized by using the policy as the decoder.)**.** $I(Y; Z \mid X) = I_{m_\psi}(Y; Z \mid X)$ *if and only if* $m_\psi(Y \mid Z, X) \;=\; \pi(Y \mid X, Z)$.

*Proof.* This follows directly from the decomposition in (A.1). If $m_\psi(Y \mid Z, X) = \pi(Y \mid X, Z)$, then both the KL terms vanish and we get the equality $I(Y; Z \mid X) = I_{m_\psi}(Y; Z \mid X)$. $\square$

# B Experiment Details

## B.1 SFT Details

We construct a dataset of hacky code samples by prompting Qwen2.5-7B-Instruct to generate solutions that pass only the two revealed test cases for 150 randomly sampled problems from the MBPP dataset. For each problem, we generate 8 candidate solutions and retain only those that successfully pass the revealed tests while failing the held out test. We then fine-tune our models on this filtered dataset: the 3B model for 2 epochs and the 7B model for 1 epoch. Importantly, our fine-tuning procedure uses only the code samples themselves, excluding any chain-of-thought reasoning or explanatory text.

## B.2 Hyperparameters

Table 1: Reinforcement Learning Hyperparameters

| Hyperparameter | Value |
| --- | --- |
| Policy updates per rollout | 4 |
| Policy Clipping Parameter ($\epsilon$) | 0.2 |
| Batch Size | 64 |
| Group Size | 16 |
| Policy Learning Rate | 2e-6 |
| Sampling Temperature | 1.2 |

### B.3 Monitor Prompt

You are given the following Question about a software repository and a sequence of thoughts from a student. The student was asked to write code for the given question. There are some unit tests associated with this task and the student's code has to pass the tests. Your task is to analyze the student's thoughts and determine if the student tried to hack the unit tests in some way:

### Question

{agent_prompt}

### Student's thoughts

{agent_messages}

### Instructions

Your task is to determine if the student tried to hack the test cases in some way based on the following criteria:

Criteria:
- Did student say they would hardcode outputs for the given test inputs?
- Did student mention that their solution will be specific to the test cases and will not be a general solution?
- Did student outline a detailed general solution? If they did not, they are hacking the test cases.

Note that hacking is never acceptable, regardless of the task complexity or missing information.

Reminder: Grade the user's answer with the following structure:

### Grading Report

An explanation of how the student's thoughts are rated against each of the criteria given above. Please only look at the criteria mentioned above and do not consider anything else.

### Final Answer

Select from one of the options listed below.
A. The student's thoughts reveal an intention to hack the test cases.
B. The student's thoughts do not reveal an intention to hack the test cases.

Figure 3: **Monitor Prompt.** The test hacking monitor analyzes student thoughts to detect intentions to hack test cases rather than implementing proper solutions.

