# OpenReview forum: "Information-Theoretic Conditions for Chain-of-Thought Monitorability and Methods for Improving It"
_NeurIPS.cc/2025/Workshop/Reliable_ML — NeurIPS 2025 - Reliable ML Workshop_

### Official Review · Reviewer_uLhZ · 2025-09-10
**Probabilistic theory on CoT monitorability**

**Rating:** 6
**Confidence:** 2

**Review:**

The author proposed a mutual information based theory on when CoT can be monitored. The author defined the monitorability as Bayes optimal estimator is accurate. The author then proved conditions when this can be achieved. Based on this observation they proposed method to improve monitorability. I am not fully familiar with then literature on CoT monitoring. But the information theory piece appears correct albeit standard. It worth being seen if such formalism was not in the literature yet.

---

### Official Review · Reviewer_NF7y · 2025-09-20
**Information-Theoretic Conditions for Chain-of-Thought Monitorability and Methods for Improving It**

**Rating:** 6
**Confidence:** 2

**Review:**

Summary: The paper studies when chain-of-thought (CoT) can be reliably monitored and how to make it easier to monitor. Using an information-theoretic lens, it argues that monitoring is only feasible if the CoT actually carries signal about the attribute of interest and is linked to the model’s final action. It then proposes training objectives that increase this linkage (including a label-free mutual-information objective) and shows, on a coding benchmark with “test-hacking” behavior, that these objectives keep CoT monitorable and reduce reward-hacking compared to standard training.

Strengths: Clear formalization of necessary conditions for CoT monitorability; a neat decomposition explaining why practical monitors can underperform the Bayes ceiling; and a simple, label-free training signal (maximize I(O;Z∣X)) that shows concrete gains on a plausible hacking environment. The setup and metrics (MI, hacking rate, monitor accuracy, true pass rate) make the story easy to follow.

Weaknesses: Generality to other tasks, attributes, or stronger adversaries isn’t fully established. The MI estimator uses group rollouts and truncation (tanh), which is reasonable but could bias learning—more ablations would help.

Suggestions: I would like to see more possible domains and datasets. It would be great if authors could add more experiments.